

# Cross-scale strain analysis in the Afar rift (East Africa) from automatic fault mapping and geodesy

Alessandro La Rosa[1,2], Pauline Gayrin[1,3], Sascha Brune[1,3], Carolina Pagli[2], Ameha A. Muluneh[4,1], Gianmaria Tortelli[5], and Derek Keir[5,6]

[1] GFZ Helmholtz Centre for Geosciences, Telegrafenberg, 14473 Potsdam, Germany
[2] Dipartimento di Scienze della Terra, Università di Pisa, Via S. Maria, 53, 56126 Pisa, Italy
[3] University of Potsdam, Potsdam, Germany
[4] Center for Marine Environmental Sciences, University of Bremen, Bremen, Germany
[5] Dipartimento di Scienze della Terra, Università degli Studi di Firenze, Via G. La Pira, 4, 50121 Florence, Italy
[6] School of Ocean and Earth Science, University of Southampton, European Way, SO16 3ZH Southampton, UK

*Correspondence to*: Alessandro La Rosa (alessandro.larosa@dst.unipi.it)

**Abstract.** The formation of continental rift systems is characterised by the interplay of magmatic and tectonic processes. Their evolution involves a wide range of time scales, from centennial scales of the seismic and diking cycles to strain localisation during millions of years of continental thinning. Our understanding of rift processes at different spatial and time scales is limited by relatively short temporal coverages of geophysical measurements and by spatially discontinuous geological datasets. Here we propose a novel method for the automatic extraction of faults and the calculation of time-averaged strains distributions using topographic information from Digital Elevation Models. We apply this method to map ~4000 individual faults within a ~70 thousand km$^2$ area of the Afar rift (East Africa), where the Nubian, Arabian and Somalian plates diverge. By comparing our results to rock dating and recent decadal geodetic measurements we deduce the rift's deformation history since 4.5 Ma and study its relationship with the current tectonic and magmatic activity. We show that the external portions of the Central Afar rift are not the mail locus of strain and rifting processes have migrated toward the axis where magma emplacement focusses strain rates due to the mechanical and thermal weakening of the crust. Increasing strains toward north-west suggest a progressive migration of the rifting process in the same direction. Conversely, Southern Afar is characterized by two systems of cross-cutting faults that respond to different strain regimes driven by the separations of the Arabian and Somalian plates from Nubia. This study demonstrates the effectiveness of our new method in quantifying fault activity and strain distribution in extensional settings and provides new insights into the spatial and temporal evolution of rifting in Afar.

**Short Summary**

We propose a new method to map faults automatically in DEMs and measure long-term crustal deformation in rift contexts. By combining our data with rock ages, we reconstruct rift evolution in Afar during the last 4.5 Myrs. We show that the rift axis is most active, with rifting propagating northwest over time. Here magma promotes crustal deformation and faulting caused by dike opening. In the southern sector Afar, two fault systems respond to different motions of diverging tectonic plates.





**1 Introduction**

35        Rifting of continents is characterized by the interplay between faulting due to the mechanical stretching of the

lithosphere and the production and intrusion of magma. These processes are highly variable across the spatial and temporal
scales. During rifting, magma and tensile strains accumulate in the lithosphere to be released by dike intrusions and/or faulting
during cycles that last $10^2$-$10^3$ years (e.g., Ebinger et al., 2010; Wright et al., 2012). The locus of extension may shift during
rifting, with progressive in-rift strain focusing or rift jumps mainly controlled by plate tectonic interactions and motions of
magma at depth (e.g., Wood, 1983; Wolfenden et al., 2004; Tortelli et al., 2022; Brune et al., 2023). As such, understanding
the evolution of continental rift systems requires investigation of magmatic and tectonic processes at a range of spatial and
temporal scales. Geodesy (e.g., InSAR and GPS) can measure the surface expressions of these processes at regional scale, with
millimetre accuracy and on decadal time-scales (e.g., Biggs et al., 2009; Metzger and Jónsson, 2014; Pagli et al., 2014; Doubre
et al., 2017; Moore et al., 2021; La Rosa et al., 2024), yet resolving the accumulation and release of strain in the lithosphere
over longer time-scales requires geological observations (Wright et al., 2012). To this aim, the analysis of faults forming in
age-constrained rocks can be used to quantify rift deformation over geological time scales (e.g., Dumont et al., 2017; Riedl et
al., 2022).

48        Fault mapping from satellite-based images allows for high-resolution investigation of regional-scale networks. Well-

preserved, surface expressions of normal faults in Digital Elevation Models (DEMs) of rifted areas can be used to retrieve the
horizontal and vertical components of extension. However, manual fault mapping at the whole rift scale can be time-consuming
and prone to interpretation biases. To overcome these limitations, various automatic or semi-automatic methods for fault
mapping have recently been developed (e.g., Zielke et al., 2010; Stewart et al., 2018; Gianpietro et al., 2021; Mattéo et al.,
2021; Scott et al., 2021). The algorithms developed by Zielke et al. (2010) and Stewart et al. (2018), for example, allow for
automatic fault detection and the measurement of lateral and vertical displacements by exploiting geomorphological offset
(e.g., rivers). Scott et al. (2021) developed a tool for the semi-automatic fault detection and scarp height estimation using length
and slope information calibrated from manually-mapped faults. Mattéo et al. (2021) instead used a U-Net Convolutional Neural
Network method to automatically detect fractures and faults in both optical imagery and DEMs.

58        Here we use the Python-based Fault Analysis Toolbox (Fatbox) originally developed by Wrona et al. (2022). Fatbox

can detect individual faults in different types of raster data, and has previously been applied to automated interpretations of
seismic datasets (Wrona et al., 2023), numerical models (Neuharth et al., 2022) and DEMs (Gayrin et al., 2023) by means of
edge detection algorithms. Here, we extended the Fatbox-based workflow for detecting normal faults in DEMs, by
implementing two workflows for the filtering of artifacts and the calculation of minimum total strains from the horizontal
components of fault. Our method was applied to the Afar rift (East Africa) to reconstruct the distribution of strain in the central
and southern sector of the rift during the last ~4.5 Myrs. The Afar rift is the ideal place for this kind of application as the arid



environment and associated low erosion rates preserve fault scarps (Dumont et al., 2017). Until now, analysis of strain accumulation and release in Afar were limited by the relatively short time-window of geodetic techniques (e.g., Ruegg et al., 1984; Calais et al., 2008; Pagli et al., 2014; Doubre et al., 2017), while the strain distribution over geological time-scales has been investigated only in some sectors of the rift (e.g., the eastern-central Afar) and at low resolution (~20 km, Polun et al., 2018). Long-term and regional strain patterns are therefore poorly understood. To address this issue, we combined measurements of minimum total strains with rock dating (Courtillot et al., 1984; Kidane et al., 2003; Lahitte et al., 2003; Feyissa et al., 2019) to first retrieve time-averaged geological rates of deformation (e.g., Riedl et al., 2022), and second to compare our results with geodetically-derived strain rates between 2014-2021 (Muluneh et al., 2024; La Rosa et al., 2024). Our approach allowed us to reconstruct the progressive migration of extension from external sectors of the rift, now inactive, toward discrete axial magmatic segments.

## 2 Geological setting

The Afar rift is a triangular depression resulting from the separation of the Arabian, Somalian, and Nubian plates (Fig. 1a). Rifting in the region started in the Gulf of Aden at ~35 Ma (Leroy et al., 2004). Rifting in Afar began approximately 30 Ma around the same time as the impingement of a mantle plume and associated eruption of flood basalts (Wolfenden et al., 2004). The Red Sea starting opening at ~19 Ma, and the northern Main Ethiopian Rift initiated at ~11 Ma (Manighetti et al., 1998; Leroy et al., 2004; Wolfenden et al., 2004, 2005; Boone et al., 2021; Rime et al., 2023). The tectonic evolution of Afar prior to ~11 Ma was hence dominated by the northeast-directed extensional regime caused by the motion of the Arabian plate (Wolfenden et al., 2004; Maestrelli et al., 2022). At ~11 Ma, the northern Main Ethiopian Rift started its northward migration, accommodating the eastward separation of Somalia in Southern Afar. At the same time a westward jump of the southern Red Sea into Afar led to the separation of the Danakil Block that caused the progressive opening of the Central and Northern sectors of Afar (Eagles et al., 2002; McClusky et al., 2010; Rime et al., 2023). Since then, the rifting process in Central Afar and Northern Afar was characterized by a progressive in-rift localization of extension. Present-day kinematic models of plate spreading from GPS measurements show the Somalian plate currently separating from Nubia at ~5 mm/yr in the ESE-WNW direction (Birhanu et al., 2016; Stamps et al., 2020). Faster spreading rates occur between Arabia and Nubia with velocities increasing from ~10 mm/year at latitude N14.5° to ~20 mm/year at latitude N12.7° causing counterclockwise rotation of the Danakil Block (McClusky et al., 2010; Viltres et al., 2020, Viltres et al., 2022).

The rift in Afar was accompanied by two major flood basalt eruptions (Tortelli et al., 2022) during ~4.5-2.5 Ma (Lower Stratoids series) and ~2.5-1.2 Ma (Upper Stratoids series) that covered almost the entire region (Fig. 1b). Recent geochemical studies suggest that these two events resulted from a north-eastward shift of strain and volcanism after a jump of the rift in Central Afar, likely induced by the inland propagation of the western Gulf of Aden branch into the Gulf of Tadjoura (Fig. 1a) between 3.0-1.7 Ma (Daoud et al., 2011; Tortelli et al., 2022). Since the Stratoids emplacement, protracted rift localization has resulted in the development of a series of en-echelon magmatic segments and tectonic grabens, which dominate



the present-day topography (Fig. 1) (Barberi and Varet, 1977; Kidane et al., 2003; Lahitte et al., 2003; Ebinger et al., 2013).
The first magmatic evidence of focused rifting dates to 1.1-0.6 Ma, with the eruption of the Gulf Basalts series more or less at
the position of the modern magmatic segments and grabens (Stab et al., 2016). The present-day magmatic segments are 50-
100 km-long and resemble mid-ocean ridge segments, with extension being mainly accommodated by dike intrusions and
faulting (Barberi and Varet, 1977; Hayward and Ebinger, 1996; Wright et al., 2012 Nobile et al., 2012; Pagli et al., 2012 Ruegg
et al., 1979; Yirgu et al., 2006; Hamling et al., 2010).
In Central Afar, the modern magmatic segments are the Assal-Goubbeth (AG), Manda-Inakir (MI) and Dabbahu-
Manda-Harraro (DMH) (Fig. 1a). The interaction between AG and MI occurs in the Makarassou area (Fig. 1) and is
characterized by NNW-striking, low-angle normal faults (Manighetti et al., 2001). To the west, the interaction between AG
and DMH occur across a ~100 km-wide area hosting the en-echelon grabens of Alol Der'Ela-Gaggade, Hanlé, Dobi, Immino
and Tendaho-Goba'Ad (Fig. 1a) (e.g., Manighetti et al., 2001; Pagli et al., 2014; Doubre et al., 2017; Moore et al., 2021;
Muluneh et al., 2024). These grabens are controlled by normal faults with throws > 200 m, and the grabens are partially filled
by Pleistocene-to-Recent deposits with varying thickness of 200-1000 m (Abbate et al., 1995). Among these grabens, Tendaho
is suggested to have been a magmatic segment with fissural eruptions between ~1.0-0.8 Ma (Fig. 1a) while minor activity after
~0.2 Ma focused at the shield volcano Dama Ali (Acocella et al., 2008; Bridges et al., 2012; Ebinger et al., 2013; Tortelli et
al., 2024). The current activity at the Central Afar grabens mainly consists of mechanical faulting with considerable seismicity
(up to Mw ~6.0) and of the emplacement of magmatic sills in the mid-to-lower crust (Hammond et al., 2011; Ruch et al., 2021;
Ahmed et al., 2022; Rime et al., 2024; La Rosa et al., 2024).
In Southern Afar, the extension following the emplacement of the Lower Stratoids led to the development of the main
Adda'Do graben at the axis of the rift, and the Karrayyu graben further northwest (Varet, 2017; Rees et al., 2023). Adda'Do
is ~100-km long, strikes NNE and is bounded on both sides by parallel half-graben structures. The graben has no evidence of
eruptive fissures and is only dissected by the main central volcano Yangudi (Rees et al., 2023). Overall in Southern Afar,
NNE-striking faults interact with a system of ~EW-striking faults, forming locally T-shaped architectures (Maestrelli et al.,
2022, 2024). Intersecting structures, combined with recent observation of an EW-striking dike that intruded in 2001 (Keir et
al., 2011) have been interpreted as caused by two long-term co-acting extensional regimes related to the separation of Arabia
and Somalia from Nubia (Maestrelli et al., 2024).

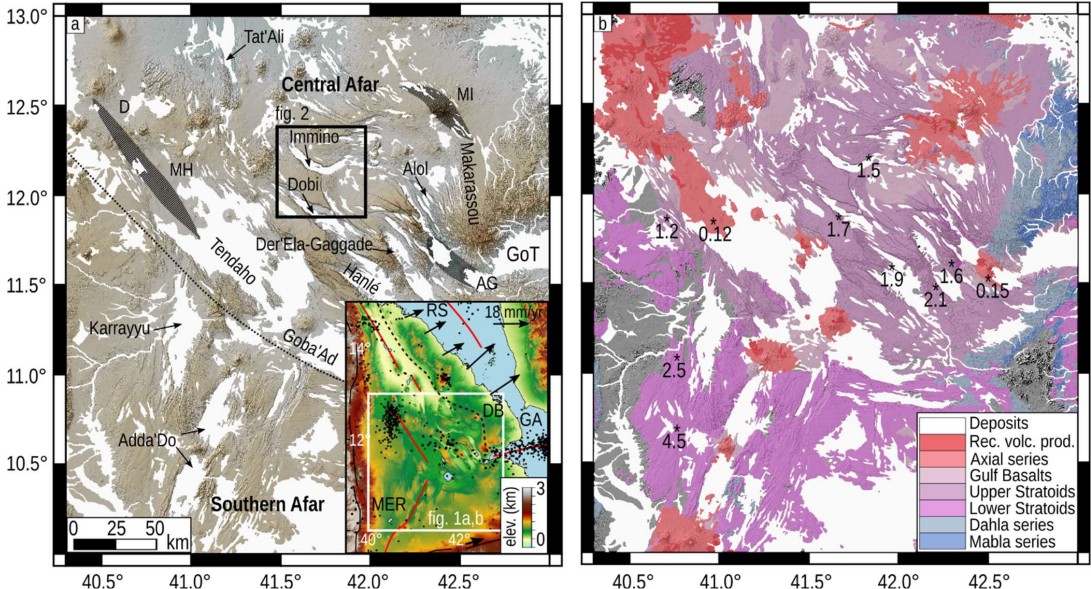

**Figure 1 -** Geological and tectonic setting of Afar. a) Topographic map, current axial magmatic segments (black dashed areas) and major grabens in the study area. The black dashed line separates the Central and Southern Afar sectors. The inset in a) shows the current plate kinematics characterizing Afar (black arrows) with red lines marking the Red Sea (RS), Gulf of Aden (GA) and Main Ethiopian Rift (MER) branches. The black dashed line in the inset marks the SW boundary of the Danakil Block (DB). D= Dabbahu, MH = Manda-Harraro, MI = Manda-Inakir, AG = Assal-Goubbeth, GoT = Gulf of Tadjoura. b) Geological map depicting the volcanic products erupted in Afar since ~ 20 Ma (Modified after Tortelli et al., 2022 and Rime et al., 2023). The values reported in b) are rock ages from Courtillot et al. (1984), Kidane et al. (2003), Lahitte et al. (2003); Feyissa et al. (2019). Areas covered by recent deposits are shown in white. The FABDEM V1-2 DEM (Hawker et al., 2022) is used as figure basis.

**3 Method**

In this study, we used an automatic fault extraction workflow implemented in the Fatbox software (Wrona et al., 2022). The methodology behind Fatbox is already provided in Gayrin et al. (2023) and Wrona et al. (2023). Here we expanded the original package for DEMs analysis by implementing new filters, and by building up a novel workflow for the analysis of deformation components, such as the horizontal extension and the second invariant of strain (Fig. 2 and Fig S1). The whole method was applied to two DEM data sets: (1) Copernicus GLO 90 m (Airbus, 2020), and (2) FABDEM V1-2 (Hawker et al., 2022), both covering an area of ~108 thousand km$^2$. The latter is a global elevation dataset obtained by filtering the contribution of buildings and trees from the Copernicus GLO 30 m DEM (Hawker et al., 2022). As a further validation, we also compared the results with those obtained from faults mapped manually in DMH.



### 3.1 Automatic fault extraction in Fatbox

Normal faults can induce sharp changes in the Earth's surface elevation due to relative vertical and horizontal motions between two crustal blocks. In the analysis of digital images, such as DEMs, these sharp gradients at adjacent pixels are referred to as *edges*. The basic rationale behind the faults extraction method in Fatbox is the detection of edges in the DEM and their subsequent conversion to a network called *graph* (Wrona et al., 2022, 2023). However, edges are not just related to faults, but also to other geological elements such as lava flows, volcanoes, and river valleys, that introduce undesired artifacts. In this study, we thus adopted a modified version of the original Fatbox workflow presented in (Wrona et al., 2023) to face the new requirements of DEMs application and artifacts filtering in Afar.

A major difference between faults and other geological elements in a DEM is constituted by elevation changes and their spatial variability. In Afar for example, lava flows are characterized by lower elevation changes and vary spatially over shorter length scales than faults. On this basis, we first applied an edge-preserving mean filter to highlight the long-range variability in the DEM, as that related to faults, while smoothing out the short-range noise caused by lava flows and other volcanic morphologies (e.g. scoria cones). This method calculates the mean of pixel values within a moving window, including just those whose absolute elevation difference exceeds a certain threshold value. For the 30 m DEM, a spatial window of 20 pixels and a threshold of 5 m allowed us to better emphasize fault scarps while smoothing surrounding noise (Fig. S1a, b).

Edge detection was performed on the filtered DEM using a Canny operator that tracks edges by hysteresis after applying an additional low-pass Gaussian filter (Fig. S1c). This is controlled by a kernel standard deviation ($\sigma$) along with two minimum and maximum gradient magnitude thresholds (Canny, 1986). These parameters are key as they influence the connection between adjacent features and affect the final fault number and segmentation. For the 30 m DEM, a Gaussian filter with $\sigma$ equal to 3.5 and minimum/maximum thresholds of 1/14 was the best combination of parameters to remove artifacts but still maintain a robust fault segmentation. Additionally, small objects with size of 50 pixels and within a neighbourhood of 10 pixels were removed from the edge raster image, while bigger residual artifacts were removed after the graph generation.

A graph was generated after skeletonization of the edges in the raster image (Wrona et al., 2023) (Fig 2b, and Fig. S1d). In the graph, each linear element representing a fault (or some undesired linear feature) is considered as an individual component and identified by an individual number. Each component is made up of nodes and connecting lines (here referred to as *fault sections*), located in a Euclidean space (Fig 2c). It follows that we can perform geometrical analyses of the graph components and, by knowing the resolution and coordinates of the initial raster, we can translate them either in a metric or a geographic reference system.

We further developed a very effective filtering operation that operates at the graph level. A geometrical parameter that helps in distinguishing between faults and artifacts is the linearity ($\Lambda$), described as the ratio between tip-to-tip ($l$) and total length ($L$) of a linear feature. Faults are commonly characterized by a higher linearity than wavy artifacts (see Fig. S1 and



Table S1). However, this assumption is not always true: As can be seen in Table S1, small linear artifacts can exist and have
higher linearity than longer curvilinear faults. To address this problem, we introduced a normalized scale-dependent linearity
filter into Fatbox that considers the area covered by linear elements. Area-based methods have been shown to be more robust
and less sensitive to noise, and various formulations of linearity with varying degrees of complexity have been proposed in the
literature (e.g., Zunic and Rosin, 2011; Zunic et al., 2016). We therefore calculate a normalized scale-dependent linearity as:
$$\Lambda_n = (l^2/L) \,/\, (L_{max} - L_{min}) \tag{1}$$
with the denominator being the normalization factor. Fig. S1 and Table S1 show how this approach successfully under-weights
the $\Lambda_n$ values of small linear artifacts while over-weighting $\Lambda_n$ values characterizing longer curvilinear faults. The filtering is
finally applied by simply removing all the graph components having $\Lambda_n$ lower than a certain threshold.

**3.2 Fault scarp quantification**

185       The extraction of the fault scarp is performed by combining the graph and the original unfiltered DEM (Fig 2d-e).

For each fault section in the graph, Fatbox calculates the mid-point and draws an orthogonal profile by extending it to a given
distance $d$, symmetrically toward the foot-wall and the hanging-wall. For each fault-orthogonal profile, the algorithm detects
the two hinge points of the fault scarp (here referred to as pick-up points) by calculating the slope percentage at adjacent
increments of the profile and stopping when its value reaches a given lower threshold (Fig 2d-e). This approach successfully
isolates just the profile portion representative of the fault scarps and automatically adapts to the progressive tip-ward reduction
of the fault throw, preventing the overlap of profiles pertaining to adjacent faults (Fig 2d-e).

193       For each profile, the algorithm calculates the throw (Fig. 2e) and the horizontal extension ($Ext_H$) as the difference in

the vertical and horizontal distances between the two pick-up points, respectively (Fig 2d). These parameters are then used to
calculate the dip angle ($\alpha$) and the displacement along the fault scarp through simple trigonometric relationships. Fault-based
measurements of $Ext_H$ were used here to retrieve the magnitude of the strain tensor (more specifically the second invariant,
$I2$). To this aim, we first derived the EW and NS components of $Ext_H$ as:
$$Ext_{EW} = Ext_H \times \cos(\varphi) \tag{2}$$
$$Ext_{NS} = Ext_H \times \sin(\varphi) \tag{3}$$
where $\varphi$ is the extension direction in radians, assuming an angle of 0° for the EW direction. We converted the pointwise
measurements of $Ext_{EW}$ and $Ext_{NS}$ to a raster grid (Fig. S2). The bin size of the raster is a key parameter as it influences the
number of measurements from adjacent faults falling within a pixel, and the correspondence between pointwise and areal
extension (Fig. S2). For our study, a bin size of 300 m resulted in each pixel being representative of measurements of individual
fault portions. To prevent averaging or summing effects caused by presence of multiple segments pertaining to the same fault
within the same pixel, we calculated the length-weighted components of extension ($WExt_{EW}$ and $WExt_{NS}$), corresponding to
the sum of products between extension components and fault section lengths, divided by the total segment length within each
pixel (Fig. S2). Fig. S2 shows the agreement between the components of pointwise and areal extension. We refined our high-



resolution dataset by manually masking pixels related to residual artifacts in the graph. The strain components $\varepsilon_{EW}$ and $\varepsilon_{NS}$
were then retrieved by dividing the components of extension for the pixel size. Finally, the second invariant of the strain tensor
was calculated following Gerya (2019) as:
$$I2 = \sqrt{1/2(\varepsilon_{EW}^2 + \varepsilon_{NS}^2)} \tag{4}$$

We finally resampled our data to a ~3 km pixel size to ease the comparison between our dense fault-based $I2$ measurements
and the regional (~15 km resolution) geodetic strain rates obtained by Muluneh et al. (2024). Resampling was performed by
directional summation of the pixels in the high-resolution maps of the $WExt_{EW}$ and $WExt_{NS}$, followed by recalculation of the
$I2$ at lower resolution (Fig S3).

216   A DEM includes contributions to the elevation by deposits and lava flows, as those observed within the major grabens

and magmatic segments in the study area. We therefore stress that our approach reconstructs only the most recent history of
slip on a fault that ruptured the uppermost geological layers. Our $I2$ measurements can hence be considered as representing
the minimum geological strain. In particular, considering the majority of faults rupturing the Upper and Lower Afar Stratoids
formations, our maps mainly depict the distribution of strain during the last 4.5 Ma, the lower age limit of the Lower Stratoids
formation (Tortelli et al., 2022).

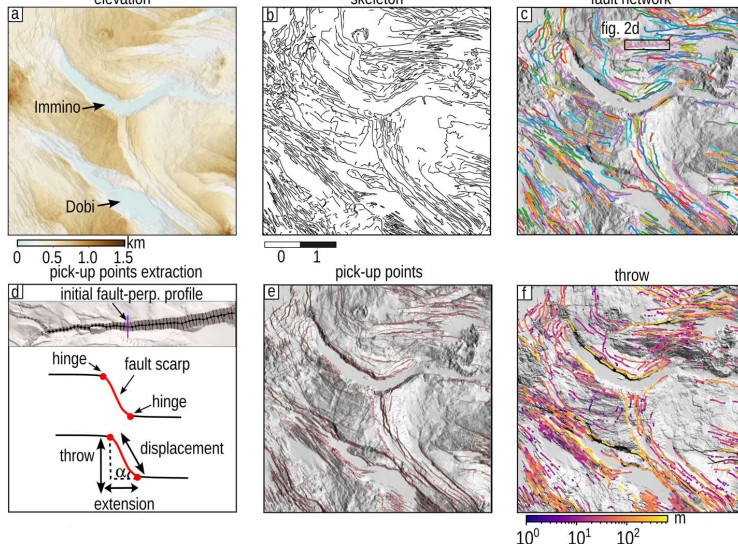


**Figure 2** – Main steps of the Fatbox workflow developed in this study. a) Hillshade of the input DEM. b) Skeletonized linear
elements identified by the edge detector, c) Filtered fault dataset converted to a graph. Colours here represent the fault
identification number. d) Basic sketch of the method for the fault scarp detection. e) Pick-up points detected every 250 m along
the fault scarp. f) Final map of along-strike throw measurements at each fault. The FABDEM V1-2 DEM (Hawker et al., 2022)
is used as figure basis.



## 4 Results

### 4.1 Fault statistics

Our automatic fault extraction workflow allowed us to map 3917 and 2165 individual faults from the 30 m (Fig. 3) and 90 m (Fig. S4) DEMs, respectively. Fault-length distributions and orientations for both datasets are presented in Fig. 3b-c and Fig S4. The datasets show similar fault orientations with a dominant NW-striking system, associated with a secondary system of ~SW-striking faults (Fig 3b and Fig S4b). The 30 m dataset consists of fault-lengths ranging between ~0.8 km and ~35 km, with the dominant class being 2.5-5.0 km, while the 90 m dataset is characterized by fault lengths between ~2.7 km and ~ 45 km, dominated by 5-10 km-long faults (Fig 3c). The histograms in Fig. 3c show that the largest differences between the two datasets are observed in length classes < 5 km, while longer faults are well represented in both datasets. We attribute these differences to the highest fault segmentation captured in the 30 m DEM (Fig. 3a and Fig. S4a). Denser fault networks have been detected for example at Makarassou, north and west of Hanlé, as well as around Immino (Fig. 3, Fig. S3c, f and Fig. S4).



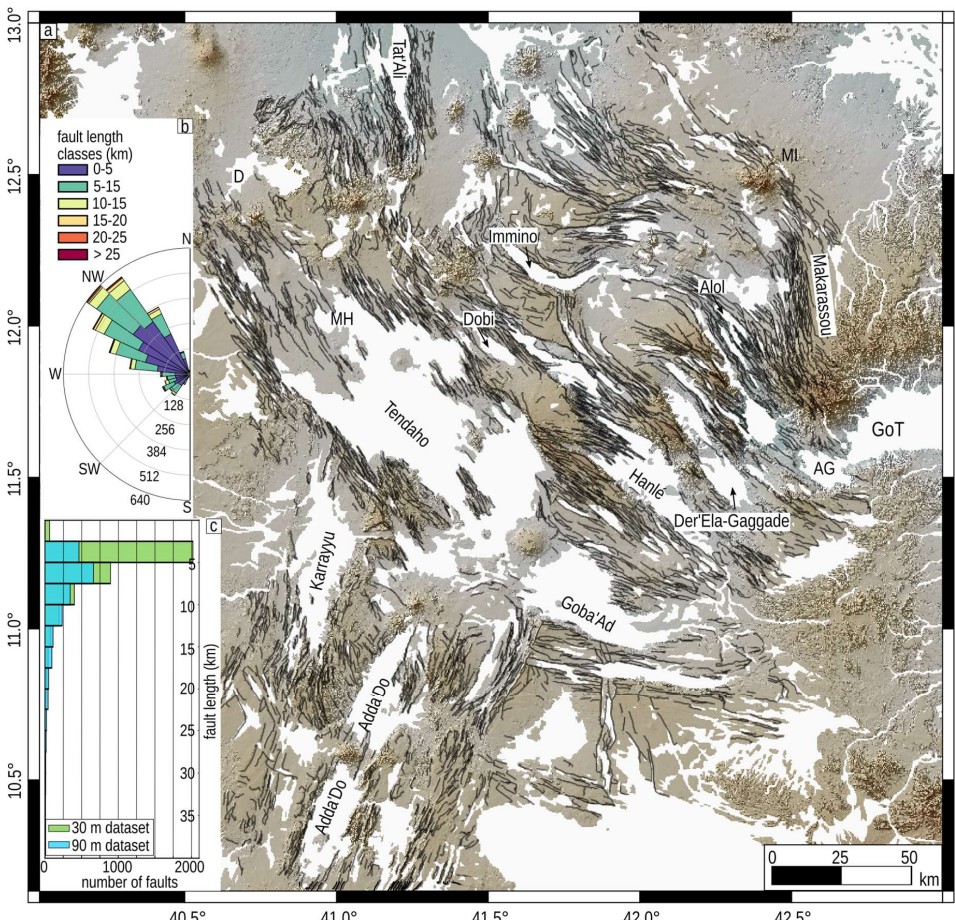

240

**Figure 3 –** Results of the automatic fault extraction with Fatbox and statistical analysis. **a)** Map of 3917 individual faults
(black lines) extracted from the 30 m FABDEM. b) Length-weighted rose diagrams of fault strike distribution for the 30 m
dataset. c) Histograms comparing the fault-length distributions obtained with a 30 m and a 90 m DEM. The histogram binning
is 2.5 km. Names and abbreviations are the same as in Figure 1. The FABDEM V1-2 DEM (Hawker et al., 2022) is used as
figure basis.

246

247

248



**4.2 Data validation**

Geological strains (*I2*) from the 30 m and 90 m datasets show the same spatial distribution with strain maxima ranging between 0 and ~0.5 (Fig. S5). Spatially scattered residuals between the two datasets range ±0.30 due to natural local differences at the scale of single pixels (Fig. S5c). While the two datasets are comparable, we based our observations and interpretation on the 30 m dataset as it provides higher detail and larger number of faults. As a further, independent validation, we compared the results obtained from the 30 m DEM with those obtained from faults mapped manually. For this test, we mapped faults in the same sector of the DMH covered by the across strike profiles in Figure 6, and calculated the associated strain (Fig. S6). We then isolated the faults and the associated strains covering the same area in the 30 m dataset. A comparison between the two datasets shows that they are characterized by similar strain values and spatial distribution (Fig. S6). Residuals between the manual and automatic datasets range ±0.1, with just an outlier pixel giving a residual of 0.15. Assuming the manual dataset being representative of 100% of the strain, we calculated that our automatic approach successfully retrieved 93.4% of the total strain.

**4.3 Spatial and temporal distribution of strain in Afar**

In this section, we compare the results of the 30 m dataset with regional geodetic strain rates derived from InSAR and GPS during 2014-2021 (Muluneh et al., 2024; La Rosa et al., 2024) to address the link between short- and long-term deformation processes during rifting. We stress that our workflow provides the distribution of regional tectonic strain accommodated by faults over millions of years. The geodetic strain on the other hand includes contributions from all deformation processes during rifting, such as magmatic activity, over decadal to annual time frames. Strain distributions are compared via a series of profiles crossing the most important tectonic and magmatic features in Central and Southern Afar, including the active magmatic segments and the major grabens (Figs. 4-6). We then converted geological strain to strain rates under the assumption of continuity of the rifting process (e.g., McClusky et al., 2010; Reilinger et al., 2011), which provides lower bounds for the geological strain rates. The comparison was possible only where well-constrained rock dating was available. Therefore, we used rock ages available close to the profiles (Courtillot et al., 1984; Kidane et al., 2003; Lahitte et al., 2003; Feyissa et al., 2019) but it was not possible to apply this approach to the entire strain map. Differences between the geological strain rates calculated at different portions of the rift provide insights into the spatial migration of the rifting process with time, while differences between measured geodetic and expected geological strain rates might suggest a possible role of magmatic processes in assisting extension. Note that the faults mapped in this study, and controlling the major tectonic grabens and magmatic segments, mainly developed in the Lower Stratoids in Southern Afar, and in the Upper Stratoids and the recent-most axial products in Central Afar (Ebinger et al., 2013; Tortelli et al., 2022; Rime et al., 2023). Our temporal reconstruction is thus limited to the last ~ 4.5 Myr in Southern Afar, and the last ~2.5 Myr in Central Afar. Figure 4 shows the results at the rift scale, while Figures 5 and 6 show details of the strain distribution at the scale of individual magmatic segments.



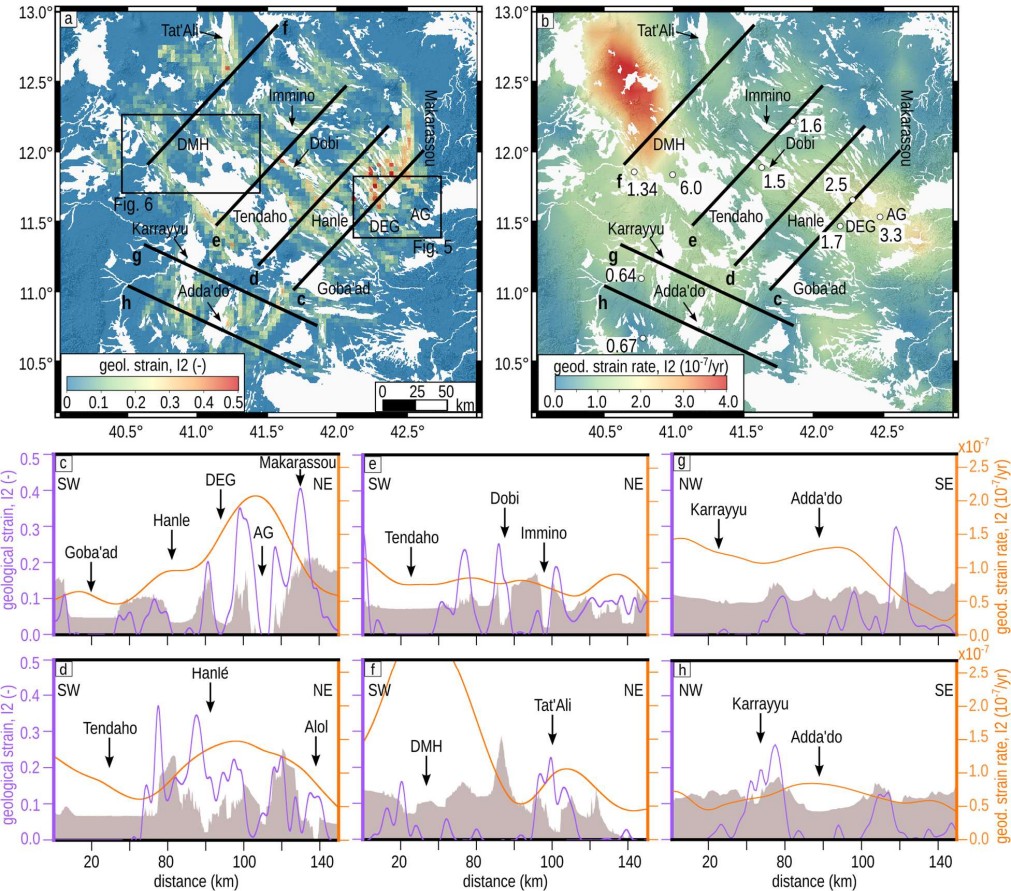

**Figure 4 –** Results of our fault-based strain analysis and comparison with deformation rates from geodesy. a) Map of the total geological strains (second invariant *I2*) at a 3 km resolution calculated from faults in Fig. 3a. (30 m dataset). Note that our fault-based analysis derived geological strain that has been accumulated since ~4.5 Myr in Southern Afar, and ~2.5 Myr in Central Afar when previous faults have been covered in flood basalts of the Upper Stratoid Series. We saturated the colour bar for values above 0.5 to better highlight the strain variability. b) Map of geodetic strain rates (second invariant) at 15 km resolution, modified after Muluneh et al. (2024). Black lines are the profile tracks shown in (c-h). The white dots and related numbers in b) represent the calculated geological strain rates (in units of $10^{-7}$/yr). c-h) Large-scale profiles comparing geological strains (purple) and geodetic strain rates (orange). The brown filled profiles show the elevation for orientation (vertically exaggerated by a factor of 1/40). The FABDEM V1-2 DEM (Hawker et al., 2022) is used as figure basis.

Geological strains ranging between 0.3 and 0.6 are observed at the AG magmatic segment and in the Makarassou area (Fig 1 and Fig, 4a, c). At AG, strains are ~0.4 at the northern tip, along faults shaping the southwestern margin of the Assal Lake graben (Fig 4 and Fig 5a, c). K/Ar dating of the top of the Upper Stratoids sampled in the area provide an age of

~1.6 Ma (Courtillot et al., 1984), which results in a long-term strain rate of ~2.5x10$^{-7}$/yr, notably similar to the geodetic strains
currently accumulating there. The axial sector of AG is represented by the Fieale area, the locus of recent-most lava flows and
the 1978 rifting episode. Here, faults developing above 0.15 Ma old volcanic products accommodated geological strains of
~0.05, corresponding to strain rates of ~3.3x10$^{-7}$/yr (Fig. 5d). Geodesy in this area measured strain accumulating at a rate of
~2.0x10$^{-7}$/yr (Muluneh et al., 2024; La Rosa et al., 2024). Conversely, geodetic measurements in the Makarassou area show
very low current strain rates (< 0.2 x 10$^{-7}$/yr).

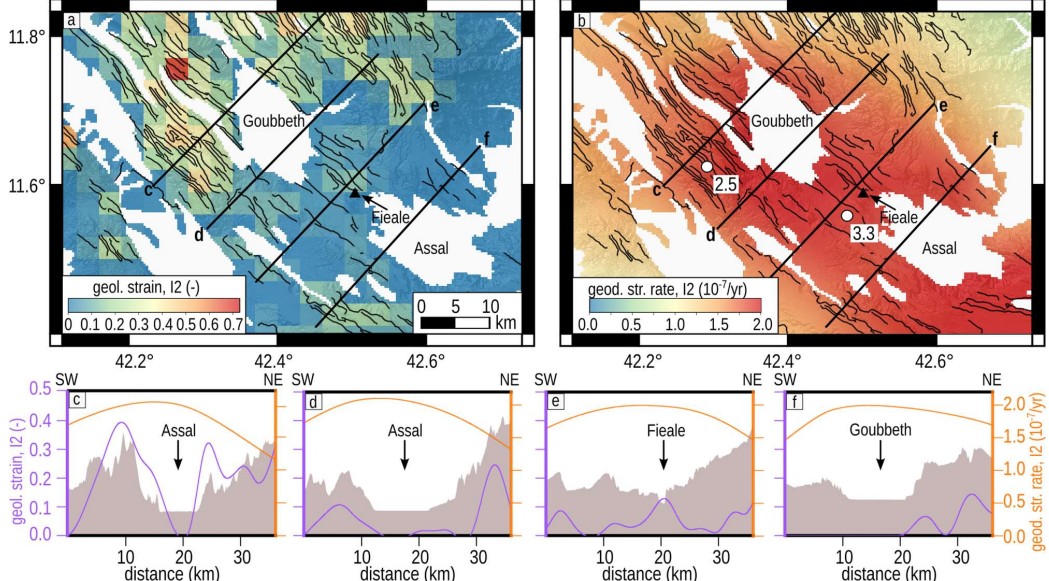


**Figure 5 -** Details of strain distribution along the Assal-Goubbeth magmatic segment. See Fig. 4a for location of focus region.
a) Map of the total geological strains (second invariant) at 3 km resolution. Map of geodetic strain rates (second invariant, I2)
at 15 km resolution, modified after Muluneh et al. (2024). Black curved lines depict faults as reported in Fig. 3a. Black straight
lines designate profile tracks shown in (c-f). Note that we modified and adapted the colour bars for a better visualization of the
strains in the area. The white dots and related numbers in b) represent the calculated geological strain rates (in units of 10$^{-7}$yr).
c-f) Detail profiles comparing geological strains (purple) and geodetic strain rates (orange). The brown filled profiles show the
elevation (vertically exaggerated by a factor of ~1/13). The FABDEM V1-2 DEM (Hawker et al., 2022) is used as figure basis.
Geological strains of 0.05-0.2 are reached along the DMH magmatic segments, the locus of the 2005-2010 rifting
episode (Fig. 4 a, f). At DMH, geological strains range from ~0.2 at margins, to ~0.05 at the segment axis (Fig.4a, f, Fig 6). In
particular, values of ~0.17 are measured on the southwestern margin of Manda-Harraro, along faults cutting the Upper Stratoids
(Fig. 6). K/Ar dating in the same area gives an Upper Stratoids age of 1.2 Ma (Lahitte et al., 2003), which result in geological
strain rates of ~1.34x10$^{-7}$/yr (Fig. 6). The inner sector of Manda-Harraro show strains of ~0.08-0.06 where axial products have
ages of ~0.12-0.04 Ma (Lahitte et al., 2003). These values correspond to geological strain rates ranging from 6.0x10$^{-7}$/yr to





15x10$^{-7}$/yr, higher than those measured at the margins and comparable to the geodetic strain rates currently accumulating in
the same area (Fig.4b, f, Fig 6). In MI, while geological strains have values up to 0.16, geodesy does not show evidence of
ongoing deformation along the magmatic segment. Finally, geological strains up to 0.4 are accommodated by faults shaping
the southern tip of TA (Fig. 4 a, f), where geodetic strain rates are still in the order of ~1.0x10$^{-7}$/yr (Fig. 4f).
Between AG and DMH, plate spreading and interaction between the two magmatic segments occur across a ~100
km-wide area of distributed extension hosting the major en-echelon grabens of Der'Ela-Gaggade, Hanlé, Tendaho, Goba'Ad,
Dobi and Immino (Fig 4 a, d-e) (Manighetti et al., 2001; Ruch et al., 2021; Rime et al., 2023; Muluneh et al., 2024). Here, the
Upper Stratoids get progressively younger toward the northwest. Average ages of ~2.0 Ma are measured between Der'Ela-
Gaggade and Hanlé (Courtillot et al., 1984) where the highest geological strains amount to ~0.35 (Fig 4e). Ages of ~1.7 Ma
have been found between Tendaho and Dobi (Kidane et al., 2003; Feyissa et al., 2019) where the highest geological strain is
~0.25 (Fig 4d). Finally, the ~1.5 Ma ages deduced at Immino (Lahitte et al., 2003) correspond to maximum geological strains
of ~0.24. These total strains and ages give approximately constant geological strain rates across the area, with values of
~1.7x10$^{-7}$/yr between Der'Ela-Gaggade and Hanlé, ~1.5x10$^{-7}$/yr between Tendaho and Dobi, and ~1.6x10$^{-7}$/yr at Immino.
However, these grabens host thick deposits (200-1000 m, Abbate et al., 1995) that hide part of the fault scarps. This implies
that the major graben-bounding faults actually accommodated higher strains during geological times. Profiles d and e in Fig.
4 show current geodetic strains cumulating at rates of ~1.5 x 10$^{-7}$/yr. Assuming an underestimation of geological strains rates
in the area, current geodetic strain rates are thus lower than the long-term ones.

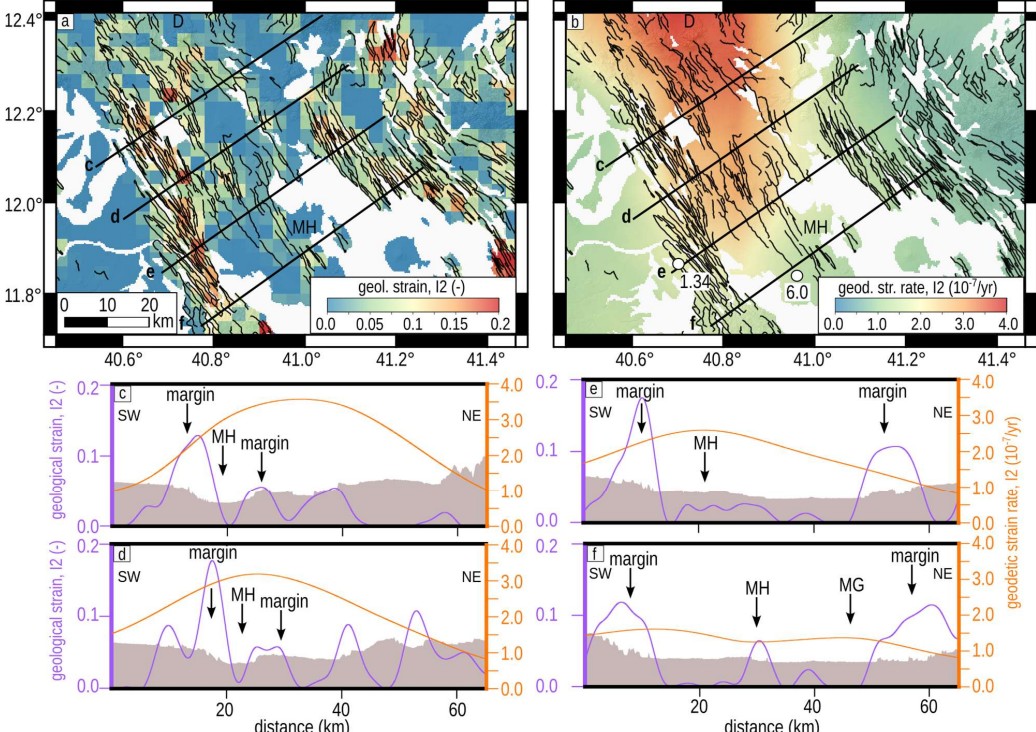

**Figure 6 -** Details of strain distribution along the Manda-Harraro (MH) magmatic segment. See Fig. 4a for location of focus region. a) Map of the total geological strains (second invariant, I$_2$) at 3 km resolution. Map of geodetic strain rates (second invariant) at 15 km resolution, modified after Muluneh et al. (2024). Black curved lines are the faults as reported in Fig. 3a. While the black straight lines are the profile tracks shown in (c-f). Note that we modified and adapted the colour bars for a better visualization of the strains in the area. The white dots and related numbers in b) are the calculated geological strain rates (in units of 10$^{-7}$/yr). D = Dabbahu. c-f) Detail profiles comparing geological strains (purple) and geodetic strain rates (orange). The brown filled profiles show the elevation (vertically exaggerated by a factor of ~1/7). The FABDEM V1-2 DEM (Hawker et al., 2022) is used as figure basis.

In Southern Afar, geological strains up to ~0.30 are accommodated at the Adda'Do graben, and along a series of parallel half-graben structures (Fig. 4a, g-h). EW-striking faults also accommodated strains up to ~0.17, shaping the northeastern boundary with the Central Afar and the internal sectors of Southern Afar. West of Adda'Do, the Karrayyu graben shows strains ranging from ~0.30 at the southern tip, to ~0.16 at the northern termination (Fig. 4a, g). The Lower Stratoids in Southern Afar vary in age from an average of ~4.5 Ma between the western margin of Adda'Do and the southern tip of Karrayyu (Lahitte et al., 2003; Feyissa et al., 2019), to ~2.5 Ma west of Karrayyu (Feyissa et al., 2019) (Fig. 1b). These values provide almost constant geological strain rates of 0.67x10$^{-7}$/yr and 0.64x10$^{-7}$/yr. Geodetic data instead show a northward increase of current strain rates from ~0.2x10$^{-7}$/y to ~1.1x10$^{-7}$/yr, with strains accumulating equally at both Adda'Do and Karrayyu. Strains are also accumulating at rates of ~ ~1.1x10$^{-7}$/yr along the northeastern border of Southern Afar.



**5. Discussion**

349
350  We analysed the spatial and temporal variations of deformation in the Central and Southern sectors of the Afar rift
351 using a method for the calculation of long-term geological strains from automatically mapped faults, and by combining these
352 observations with current geodetic strain rates and rock dating. This automatic approach was tested on DEMs at various
353 resolutions and validated with manually mapped faults. Our method successfully isolated most of the faults in the study area,
354 minimizing artifacts introduced by other morphologies in the DEMs. We show that our results are independent of the DEM
355 resolution as strains obtained from both the 30 m and the 90 m datasets are comparable with major regional faults being equally
356 represented and different fault systems being likewise well resolved. Some natural differences in the number of faults were
357 observed due to the different levels of detail and fault segmentation dependent on the DEMs resolution. Other minor differences
358 in the final strain maps are also introduced during raster resampling, yet this parameter can be changed easily and adapted for
359 analyses at different spatial scales. For our purposes, the 30 m DEM and a map resampling at 3 km provided the best result in
360 terms of number of faults detected, level of detail and similarity with independent geodetic data. A comparison with manual
361 mapping in the key area of DMH shows that our approach can resolve 93.4% of the total strain in a study area lending further
362 robustness to our automatic fault extraction techniques. In conjunction with an independent prior study (Wrona et al., 2023)
363 we conclude that Fatbox can be successfully applied to extract faults and quantify along-strike geometries from different types
364 of data. Further applications of these methods could be used to analyse analogue and numerical models that are comparable to
365 larger scale DEMs. However, we also underline that some limitations in our method remain. Our approach works best in
366 extensional tectonic settings (such as the Afar rift) where faults are active and/or characterized by sharp edges in the
367 topography. Furthermore, our method cannot resolve the strike-slip component of faults as this motion does not generate any
368 measurable vertical throw in a DEM. Further developments of Fatbox might account for the analysis of optical imagery or the
369 automatic detection and measurement of lateral geomorphological offset (e.g., rivers) or other displaced objects (e.g., edges of
370 lava flows).

371  The fault distribution, and the analysis of geological and geodetic strains combined with dated rocks shed light on the
372 long-term tectonic evolution of different sectors of the Afar rift, including individual magmatic segments and their interaction
373 zones. At the regional scale, the major fault systems and the highest strains were measured in Central Afar (Figs. 3 and 4)
374 characterized by faster spreading velocities (up to ~20 mm/yr) compared to the Southern Afar, during the last ~2.5 Myr (e.g.,
375 Doubre et al., 2017; Viltres et al., 2020). The Central Afar is characterized by a dominant system of NW-striking normal faults,
376 mainly developing along the current regional axis of the rift systems, and a secondary system of NNW-striking faults observed
377 at external sectors (e.g., in Makarrassou). Faults at the external sectors of the rift show geological strains comparable to those
378 measured at the current axis (e.g., Fig.4a, c), indicating they accommodated large extension in the past, but they are currently
379 accumulating very low strain rates. Similarly, relatively high geological strains were accommodated in MI, which is currently
380 characterized by geodetic strain rates close to zero and lack of seismicity (Muluneh et al., 2024; La Rosa et al., 2024). At the
381 current rift axis, the faults that shape the active magmatic segments and the major grabens in Central Afar cut the Upper





Stratoids and the younger axial products. The faults on the Stratoids show a progressive decrease of geological strains (and
inferred strain rates) from southeast around AG, to northwest around DMH. Our strain measurements in AG agree with the
relative high strains previously measured in the same area by Polun et al. (2018), yet some discrepancies are observed
elsewhere in MI and Makarassou. These differences might be partially caused by the different spatial resolutions characterizing
the two studies. Our observations are consistent with modern reconstructions of Afar evolution whereby the opening of
Central/Northern Afar was accompanied by the counter-clockwise rotation of the Danakil block and the progressive focusing
of extension and magmatic activity to the axis since ~1.1 Ma (e.g., Eagles et al., 2002; McClusky et al., 2010; Ebinger et al.,
2013; Viltres et al., 2020; Rime et al., 2023). In this scenario, the larger geological strain rates at AG compared to DMH agree
with faster angular velocities at the southern termination of the Danakil block (McClusky et al., 2010; Viltres et al., 2020). The
Makarassou area and the MI segment currently do not accommodate significant extension following the migration of the
extension and magmatic activity toward the rift axis. Furthermore, larger total strains in the south-eastern sector, combined
with the observation of progressive younger Stratoids toward the northwest, point toward a long-term evolution of Central
Afar characterized by a northward propagation of the rifting process, as already hypothesized by Lahitte et al., (2003) and
Rime et al. (2023).
Both AG and DMH magmatic segments show a strong increase in strain rates at inner sectors compared to the
margins. At the inner sectors we measured strains in the ~0.12 Ma-old axial products, yet the first phases of axial magmatism
are represented by the older and sparse Gulf series (~0.6 Ma). In AG, geological and geodetic strain rates are highest in Fieale.
Here, strains accommodated by dike intrusions and faulting during the rifting episode of 1978 were three orders of magnitude
higher than those currently accumulating (Tarantola et al., 1979). Opening velocities were still 3-4 times larger than average
plate spreading rates during the ~8 years following the intrusion (Ruegg et al., 1984), with an accelerated post-rifting phase
likely lasting until 2001 (Doubre et al., 2007). Geodesy shows that the ongoing inter-rifting strains are the results of a
combination of tectonic and magmatic processes beneath AG (Muluneh et al., 2024; La Rosa et al., 2024). DMH shares similar
features, with increased geological and geodetic strain rates at the inner portion of the magmatic segment. This area
accommodated strain of the order of $10^{-5}$ during the co-rifting episode of 2005-2010 (Grandin et al., 2010). The rifting episode
was characterized by thirteen dike intrusions with associated faulting, and by increased fault slip rates due to further magma
motions during inter-diking phases (Grandin et al., 2010; Dumont et al., 2016; Hamling et al., 2010; Wright et al., 2012; Pagli
et al., 2014). Recent geodetic surveys also showed that DMH is currently experiencing a post-rifting phase accompanied by
renewed magma inflow and inflation (Moore et al., 2021). The presence of a long-lived magmatic system at the AG and DMH
magmatic segments is widely testified by surface evidence of volcano-tectonic activity (Dumont et al., 2017), geochemical
reconstructions (Medynski et al., 2015; Tortelli et al., 2024) and geophysical observations (e.g., Doubre and Peltzer, 2007;
Keir et al., 2009; Wright et al., 2012; Smittarello et al., 2016). The increase of strain rates from the margin to the intruded inner
portion of the magmatic segments during geological times, together with the decadal scale increase in geodetic strain rates
during diking episodes suggest that magmatism is an important long-term process promoting strain. Numerical models of



magma intrusions in the shallow crust show that the increased heat flow and thermal gradients weaken the crust surrounding
magma bodies and induce thermo-mechanical heterogeneities that promote strain focusing, increased strain rates and fault slip
(e.g., Regenauer-Lieb et al., 2008; Douglas et al., 2016; Dumont et al., 2017; Brune et al., 2023). Shallow faulting is also
promoted mechanically by dikes as dilation induces tensile forces at the upper tips of the intrusion (e.g., Trippanera et al.,
2015). The long-term increase in the strain rates at the inner portion of the magmatic segments during at least the last 0.12 Myr
might thus have two and possibly concomitant causes. On one hand, the faults that we mapped at might be the surface
expression of past rifting (or single diking) episodes that released large strains in narrow zones during short time-spans. On
the other hand, the presence of long-lived shallow magmatic systems might have promoted a long-term increase in the fault
slip rate in a weakened crust.
A ~100 km$^2$-wide area of distributed faulting accommodates the extension between AG and DMH magmatic
segments. In this area, most of the strain focuses at Der'Ela-Gaggade, Hanlé, Tendaho, Goba'Ad, Dobi and Immino grabens
(Fig 4a, b, e, d). Here, a thick deposit cover (1000-1600 m) reduces our fault-based measurements but including the vertical
extent of the accommodation space in the throw calculation provides strains comparable to those accommodated at the adjacent
magmatic segments. Geodesy shows instead that current strain rates are accumulating at rates lower than the magmatic
segments. A possible explanation could be that some of these grabens grew by a combination of faulting and dike intrusion in
the past, as shown by the presence of axial products and volcanic centres in the southern Tendaho graben (Fig. 1).  Current
seismic imaging, geodetic modelling and crustal balance calculations instead show the presence of a lower crust heavily
intruded by mafic rocks that constitute up to the 40% of the total volume (e.g., Hammond et al., 2011; Ahmed et al., 2022;
Rime et al., 2024; La Rosa et al., 2024). Therefore, diffuse and protracted magma addition into the lower crust likely contributes
to extension and reduces fault related extension (Rime et al., 2024).
In Southern Afar, SSW-striking fault systems accommodate most of the geological strains induced by the ESE-
directed separation of the Somalian plate from Nubia. Geological strain rates are uniformly distributed and generally lower
than those measured in Central Afar. This is not unexpected considering the low spreading velocities characterizing Southern
Afar (Birhanu et al., 2016; Doubre et al., 2017). Lower strain rates were also accommodated by systems of EW-striking faults
that formed at the northeastern boundary with Central Afar and within the internal sectors of Southern Afar (Fig. 4). These
structures, along with the opening of an EW-striking dike in 2001 (Keir et al., 2011), have been recently suggested to be the
evidence of two NE-directed and EW-directed co-acting extensional regimes in the area (Maestrelli et al., 2024). Our geodetic
observations of strain rates accumulating along both EW-striking and SSW-striking faults, support this hypothesis and indicate
that Southern Afar is currently experiencing the superposing effects of the separation of both Arabia and Somalia from Nubia.
**6. Conclusion**
In this study we developed an automatic approach for the accurate identification of faults in DEMs and the calculation
of geometrical parameters, slip components and horizontal strains. Our method was tested at various DEM resolutions and the



data was validated by comparison with those obtained on manually mapped faults, yielding overall similar results. We applied
our method on the central and southern sectors of the Afar rift, where we combined fault-based measurements with geodesy
and rock dating to reconstruct the long-term distribution of strain and its relationship with the tectonic and magmatic evolution
of the rifting process. Our main results can be summarized as follow: 1) During the last ~2.5 Ma, the rifting process in Central
Afar propagated toward the NNW as suggested by higher total strain in the south-eastern sector and by the Stratoids ages
getting younger in the same direction 2) During the last 0.6 Ma, the external sectors of the Afar rift, including Makarassou and
MI are not the main locus of strain following the migration and focusing of extension to the rift axis. 3) At the magmatic
segments, strains in the inner sectors are higher than those accommodated at the margins, likely due to magmatic activity that
promotes dike-assisted extension and associated faulting. Diking also weakens crustal rheology that increases the strain rates.
4) AG and DMH interact across a ~100 km-wide area where spatially constant strains, as high as those accommodated at AG,
were uniformly accommodated during the last 2.5 Ma by a series of en-echelon grabens; 5) During the last 4.5 Myr, Southern
Afar is experiencing a bi-directional extension induced by the separation of both Somalia and Arabia from Nubia; 6) Strain in
Southern Afar is accommodated not just at the axis but also by lateral graben structures, such as Karrayyu.
**Code Availability**
The codes created in this study are provided through the Open Science Framework (OSF) repository
(https://doi.org/10.17605/OSF.IO/AV5WD, La Rosa et al., 2025). The Fatbox software is open source and can be downloaded
from GitHub at https://github.com/thilowrona/fatbox.
**Data Availability**
The data created in this study are provided through the Open Science Framework (OSF) repository
(https://doi.org/10.17605/OSF.IO/AV5WD, La Rosa et al., 2025). The 90 m resolution Copernicus DEM GLO90 is publicly
available and can be downloaded from the Copernicus Browser (https://dataspace.copernicus.eu/explore-data/data-
collections/copernicus-contributing-missions/collections-description/COP-DEM). The 30 m resolution Forest And Buildings
removed Copernicus DEM (FABDEM V1-2 DEM, Hawker et al., 2022) is publicly available and can be downloaded from the
University of Bristol webpage at the following link: https://data.bris.ac.uk/data/dataset/s5hqmjcdj8yo2ibzi9b4ew3sn. The
figures in this study have been generated using Python3 and QGIS.






**Author contribution**

ALR, SB and CP: conceptualization, methodology, formal analysis, funding acquisition. ALR, SB and PG: software, methodology, formal analysis. ALR, PG, SB, CP, AM, GT, DK: investigation, visualization, and writing (manuscript draft preparation).

**Competing interests**

The authors declare that they have no conflict of interest.

**Financial Support**

ALR acknowledges the Helmholtz Information & Data Science Academy (HIDA) for providing financial support enabling short-term research stay at GFZ-Potsdam to build-up the workflow and analyse the dataset. CP and ALR acknowledge the SpaceItUp project funded by ASI and MiUR, contract n.2024-5-E.0—CUP n.I53D24000060005. PG has been funded by the German Science Foundation (DFG) (Project No. 460760884). SB acknowledges funding from the European Union (ERC, EMERGE, 101087245). AM has been funded by German Research Foundation (DF G 537025018). DK acknowledges funding from the 2017 PRIN project-protocol MIUR: 2017P9AT72 PE10.

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
