# Peer review of "Cross-scale strain analysis in the Afar rift (East Africa) from automatic fault mapping and geodesy"

_EGUsphere, 2025_

## Referee Comment (RC1)

**General comments**

Overall, the manuscript is of very high quality. The methodology is novel, seems robust, and could be applied to many other study cases. The results from the Afar region are globally relevant and allow to better understand the late phase of rifting. One of the main interests of the paper is that it links different scales of time, liking processes happening over millions of years with processes happening over decades or even shorter. The paper is very well written and has a very clear and coherent structure. The figures and supplementary figures look good and are clear. Except for one aspect to be discussed, the results support the conclusions.

I mainly have minor comments. Some uncertainties of the methodology could be discussed in more details, even though they won't change the conclusions. I also wonder why you didn't calculate more geological strain rates from existing datings. This parameter is very interesting and allows a direct comparison with Recent geodetic data. Finally, I question one point of the conclusion regarding Manda Inakir which should be discussed and argued before being presented in the conclusion, or be toned down.

I also made several suggestions that do not relate to scientific quality, but to style or clarity, which is subjective and personal. They are just suggestions, don't hesitate to not follow them without justification.

**Specific comments**

**l.88-90:** "*Faster spreading rates occur between Arabia and Nubia with velocities increasing from ~10 mm/year at latitude N14.5° to ~20 mm/year at latitude N12.7° causing counterclockwise rotation of the Danakil Block*". This sentence is a bit confusing and not 100% correct. 10 mm/yr does not represent the velocities between Arabia and Nubia at 14.5°N, but the velocity between the Nubian plate and the Arrata Microplate (or Danakil Block). The full velocity between Nubia and Arabia is approx. 17 mm/yr at this latitude. But I think part of the problem comes from the fact that you did not yet introduce the Danakil block in your manuscript at that point. It is thus difficult to explain these velocities without the prior explanation of this concept. In my opinion, this is important to make clear because it is relevant for one of your points of conclusion. I think this concept would also be easier to explain with a bigger context map which leads me to my next point.

**Figure 1:** The figure is great but I think the insert with the full map of the Afar is quite small. It makes it even difficult to see the rotation of the Danakil Block. One possibility (but that's just a suggestion) would be to make a full standalone figure just with the context maps of the Afar Depression, allowing you to zoom out, better show the kinematics of the region

and the relation to the Red Sea and Gulf of Aden. Another possibility (again, just a suggestion), would be to just zoom out the existing insert, maybe leaving out the seismicity to not over-clutter the map.

**L. 189-194 and methods:** A question about methodology here. You assume that the horizontal distance between the two pick-up points directly corresponds to the extension. You mention that this method works well with lateral variations towards the tips of the fault. What about the influence of surface processes? Erosion of the footwall and deposition in the hangingwall will artificially reduce the angle α, increasing the measured heave (and maybe decreasing slightly the measured throw). If you restrict your measurement to the steepest part, you might alleviate this bias on the heave, but you would artificially reduce the throw. Did you check this? A good way would be to look at the distribution of α values. If they are mostly above 60°, that would be a good sign. If a lot of them are below, that might indicate a bias. Regardless of the distribution of α values, I think it would be good to mention this source of uncertainty, how you managed it and your assessment of its impact on the results.

Another point on the methods: I see that the process fails to identify very large faults in the southern part of the Immino Graben (around 12.13N, 41.69E and 12.17N, 41.88E). Looking at it on satellite images, it seems clear that these are faults (with very large throws of approx. 600m and 200m, respectively). In your opinion, what is causing this? It might again not be bad to discuss this source of uncertainty and its impact on the results.

**Figure 3:** I am a bit surprised by the rose-diagram of fault strike (b). There is this big NW trend coherent with the main orientation of the Afar rift. The second WSW to SW trend surprises me. If you look at the southern Afar (Karrayyu, Adda'Do) on the map, the faults rather seem to trend SSW. And there is almost nothing in this sector of your rose diagram. This effect is even stronger on the rose diagram of figure S4. I'm also surprised you don't have more of the NNW orientations which you can observe on the map in Tat'Ali, Makarassou (and which are more present in the rose diagram of Polun et al., 2018). I did not do any quantitative assessment of it, but it "looks" strange to me. Could you check upon this? Just to make sure there was not a small mistake in the algorithm calculating the azimuths or plotting the rose diagram.

**L. 270-272: rates.** *"The comparison was possible only where well-constrained rock dating was available. Therefore, we used rock ages available close to the profiles (Courtillot et al., 1984; Kidane et al., 2003; Lahitte et al., 2003; Feyissa et al., 2019) but it was not possible to apply this approach to the entire strain map."* I don't really understand why you didn't calculate more points. We see only 9 points on fig. 4b. There are much more datings available, from the sources you mention and from others (e.g. Kidane et al., 2003, Courtillot et al., 1984, Manighetti et al., 1998, Zumbo et al., 1995 [I have a shapefile with a compilation (and corrected coordinates) here if you want to check: https://doi.org/10.5281/zenodo.7410073 ]). I guess it would not be difficult for you

to calculate geological strain rates for more points where datings are available. These strain rates are very interesting and allow a better comparison with Recent geodetic strain rate, and they would allow to strengthen considerably your interpretation. But maybe there is a good reason why you only restricted this quantification to a few points?

**L. 426:** Where does this 1000-1600 m value come from? Previously, you mentioned 200-1000 m from Abbate et al. (1995).

**L. 452-453:** *"During the last 0.6 Ma, the external sectors of the Afar rift, including Makarassou and MI are not the main locus of strain following the migration and focusing of extension to the rift axis."* I think that this conclusion point deserves more discussion, in particular concerning the Makarassou and Manda Inakir region. I was also always intrigued by this area and by this process of strain focusing. In Manda Inakir, the strain is not huge, but it's still clearly faulted. However, the volcanic rocks there are quite Recent. Lahitte et al. (2003) and Manighetti et al. (1998) show age <0.6 Ma, down to 0.03 Ma. So, what is the geological strain **rate**? I'd guess that with these recent ages, it's not that low. This would mean it was still a locus of strain in the last 0.6 Ma.

Let's take another example: the West Harak graben (12.67N, 41.09E ). It is also very recent (0.62-0.07 Ma, Lahitte et al., 2003), heavily faulted and accommodates significant strain according to your results. So, I'd guess that the geological strain rate is also quite high. This suggests, again (maybe even more clearly than MI), that the switch to Dabbahu - Manda Hararo happened more recently than 0.6 Ma.

One could even hypothesize that these peripheral regions became inactive very recently (100s, 1000s, 10000s of years) [but that's a wet-finger guess]. And this would have important implications. It would mean that this change is not necessarily representative of a long-term trend of strain localisation at long-lived and final rift segments, but rather a short-term alternation between different areas of strain accommodation. And that these areas of strain accommodation might have changed in the past and might change again in the future. At larger time and spatial scale, the Afar region has already seen multiple rift axis jumps and abandonments. So, I'm wondering whether what we see here is a (final) rift localisation, or just one rift jump among many others.

Without quantification, this is difficult to answer. And this brings me back to my comment above: why didn't you calculate more geological strain rates points? With more points in Makarassou, Manda Inakir, but also elsewhere in Afar, you could significantly strengthen your point. With only 9 data points, it's hard to be very conclusive.

But I'm probably falling down a rabbit hole, and it's not the main point of the paper. Material for further reflections in the future.

To sum up this comment, I think you should discuss this topic more explicitly in the discussion, taking into account the aspects mentioned above, before presenting it in the

conclusion. Or just mention what we know for sure: the MI (and other peripheral regions) currently does not accommodate extension, but we don't know exactly when this change happened.

**Technical corrections**

**L. 21:** mail → main?

**L. 20-21-22:** Maybe separate in two sentences for clarity? ...*locus of strain. Rifting processes*... Just a suggestion.

**L. 91:** Maybe a small modification like "*The rifting stage in the Afar was accompanied by two further flood basalt*..." to make clear that these are not the initial flood basalt event. Just a suggestion.

**L. 95:** maybe you can also cite Tesfay et al., 2003 https://doi.org/10.1130/B25149.1bv (just a suggestion).

**L.116:** I think Varet's book was published in 2018.

**L. 260:** 93% is a very good result! Your algorithm is doing a very good job.

**Figure 4:** A nice-to-have (but not needed) improvement of your figure could be the addition of your calculated geological strain rates (white dots on b) projected on the cross-sections (c-h). It would help to link both datasets and showcase these nice results a bit better. Just a suggestion.

**L. 390:** "... *faster angular velocity*..." *faster velocity* rather, no? The angular velocity is constant but the further away from the Euler pole you are, the faster absolute velocity is.

**L.420:** "*On one hand, the faults that we mapped at might*..." typo.

**L.431-433:** If you want, you might want to add Wang et al. (2021) https://doi.org/10.1016/j.tecto.2021.228857 also showing this interpretation. Just a suggestion.

If there is anything unclear in my review or if you have any question, don't hesitate to post a quick message in the discussion.

Best regards,

Valentin Rime

---

## Author Comment (AC1)

We thank the Editor and the two Referees for their constructive comments on our study. We performed the tests suggested by Referee1 and incorporated all suggestions from both Referees in the revised version of the manuscript. In particular, we conducted new tests on the distribution of fault dip angles, their relationship with fault throws and their influence on the measured strain. We also included new constraints on rock ages for other sectors of the study area, along with new insights on the evolution of rifting and faulting recently published in literature. We believe that these changes strengthen and improve our analysis and we hope that the manuscript now meets the high standards of *Solid Earth*.

In the following rebuttal, we addressed the major comments from Referee1 and 2, while minor suggestions have been incorporated straight to the text. Similar comments from both referees have been addressed together.

**Referee 1, Comment 1**
l.88-90: "Faster spreading rates occur between Arabia and Nubia with velocities increasing from ~10 mm/year at latitude N14.5° to ~20 mm/year at latitude N12.7° causing counterclockwise rotation of the Danakil Block". This sentence is a bit confusing and not 100% correct. 10 mm/yr does not represent the velocities between Arabia and Nubia at 14.5°N, but the velocity between the Nubian plate and the Arrata Microplate (or Danakil Block). The full velocity between Nubia and Arabia is approx. 17 mm/yr at this latitude. But I think part of the problem comes from the fact that you did not yet introduce the Danakil block in your manuscript at that point. It is thus difficult to explain these velocities without the prior explanation of this concept. In my opinion, this is important to make clear because it is relevant for one of your points of conclusion. I think this concept would also be easier to explain with a bigger context map which leads me to my next point.
**Response**
As suggested, we rephrased the sentence and modified it as follows: *Present-day kinematic models of plate spreading from GPS measurements show the Somalian plate currently separating from Nubia at ~5 mm/yr in the ESE-WNW direction (Birhanu et al., 2016; Stamps et al., 2020), while the NE-directed separation of Arabia from Nubia occurs at rates of ~18 mm/yr (Viltres et al., 2022). In-between Arabia and Nubia, the Danakil block separates from Nubia with velocity increasing from ~10 mm/year at latitude N14.5° to ~18 mm/year at latitude N12.7°, exhibiting a counterclockwise rotation (McClusky et al., 2010; Viltres et al., 2020; , Viltres et al., 2022).".*

**Referee 1, Comment 2**
Figure 1: The figure is great but I think the insert with the full map of the Afar is quite small.  It makes it even difficult to see the rotation of the Danakil Block. One possibility (but that's just a suggestion) would be to make a full standalone figure just with the context maps of the Afar Depression, allowing you to zoom out, better show the kinematics of the region and the relation to the Red Sea and Gulf of Aden. Another possibility (again, just a suggestion), would be to just zoom out the existing insert, maybe leaving out the seismicity to not over-clutter the map.
**Referee 2, Comment 9**
Line 125. Why is the MH magmatic segment shown with that strange elliptic shape, whereas the others are drawn with much more detail? Uniform.
**Response**
We modified Figure 1 as suggested by both referees. We zoomed out the inset in Fig. 1a and modified the polygon marking Dabbahu-Manda-Harraro.

**Referee 1, Comment 3**
L. 189-194 and methods: A question about methodology here. You assume that the horizontal distance between the two pick-up points directly corresponds to the extension.  You mention that this method works well with lateral variations towards the tips of the fault. What about the influence of surface processes? Erosion of the footwall and deposition in the hangingwall will artificially reduce

the angle α, increasing the measured heave (and maybe decreasing slightly the measured throw). If you restrict your measurement to the steepest part, you might alleviate this bias on the heave, but you would artificially reduce the throw. Did you check this? A good way would be to look at the distribution of α values. If they are mostly above 60°, that would be a good sign. If a lot of them are below, that might indicate a bias. Regardless of the distribution of α values, I think it would be good to mention this source of uncertainty, how you managed it and your assessment of its impact on the results. Another point on the methods: I see that the process fails to identify very large faults in the southern part of the Immino Graben (around 12.13N, 41.69E and 12.17N, 41.88E). Looking at it on satellite images, it seems clear that these are faults (with very large throws of approx. 600m and 200m, respectively). In your opinion, what is causing this? It might again not be bad to discuss this source of uncertainty and its impact on the results.

**Referee 2, Comment 12**
Line 365 and following. I would mention here that, besides being characterized by ongoing extension, the area is also particularly favorable for this analysis as the dominant lithology (basalts) allows the development and preservation of sharp fault scarps. In other parts of the EARS this may not be the case as sediments or volcano-clastic deposits may result in less pronounced (and preserved) fault morphologies.

**Response**
We agree that erosion, but also fault morphologies can impact on the dip angles and measured strain. We thus explored further the distribution of dip angles, with a focus on the relationship with the fault throws, as suggested. The results of this analysis have been included in the new supplementary Fig. S7. We plotted the distribution of dip values in map view (Fig. S7a), along with the average fault dip angles against fault throw units (Fig. S7b). We found that for throws above 200m the average and maximum dip angles are consistent and mostly 20°-40° and ~60°, respectively (Fig. S7a,b), similar to the dip angles measured in the field in various sectors of the rift (e.g., Makarassou, Geoffroy et al., 2014; west of Dabbahu, Stab et al., 2016), as also with modeling of coseismic ruptures (e.g., La Rosa et al., 2019).

However, we also found that smaller faults with throws less than 200m have lower average dip angles which also decreases with decreasing throw (Fig. S7b). Visual inspection of the DEM shows this is in part due to some erosion on the external (older) sectors of the rift, but the majority of small offset faults are associated with a low dip angle monocline caused by elastic flexure of rock above a propagating fault, as also observed by Hoffmann et al. (2024) in Dabbahu-Manda-Harraro.

We tested the effect of the low dip angle of the minor faults on the total strain by fixing a dip angle of 30° to faults with throws lower than 200m and dip angle lower than 20°(Fig. S7c,d and f). The results show that changing the dip angle the spatial distribution of strain remains similar, with some decrease mainly observed on minor faults and on the external sectors of the rift (Fig. S7e,f). We now included this analysis and described the uncertainties associated with recent or possibly eroded smaller faults at lines 264-277 of the revised manuscript. Nevertheless, our analysis indicates that faults above 60° are rare in Afar and this does not necessarily indicate a bias. Low angle faulting in Afar has been observed in the field and explained with fault reactivation, pre-existing crustal weaknesses and tilted blocks for protracted extension that cause faults to have dip angles much lower than 60° (e.g., Stab et al. 2016). Previous studies of strain distribution dealt with dip angles in various ways: Assuming no major effects of erosion (e.g., Dumont et al., 2017), assuming constant dip angles of 60° (e.g., Gupta and Sholtz, 2000) or even higher (e.g., Polun et al., 2018). While an assumption on constant dip angles can work on local studies, this can cause biased strain measurements if applied at a broader scale. We thus preferred to explore the full range of possible dip angles and now better discuss the source of uncertainties, as suggested by the referee.

We also point out that, for major faults, possible local effects of erosion remain considered during fault extraction: Eroded portions of the faults generate small irregular edges that are removed by the

scale-dependent linearity filter before calculating displacement parameters. This means that heavily eroded major faults do not contribute to the measured extension. An example is just the southern part of the Immino graben mentioned by the referee. As can be seen in Fig. 2b, the skeletonized map shows that edges are correctly detected in that part of Immino, but the erosion caused the edges to be strongly nonlinear and they were removed (Fig. 2c). We report other examples of excluded eroded faults in figure 1 of this rebuttal. If eroded faults remain in the dataset, they generate local outliers that we do not consider in the analysis. This is the case of a fault close to Lake Assal (red line in figure 1 of this rebuttal) that generated a strain outlier, already reported in Fig. S5.

[Figure]

**Figure 1** - Details of the fault map extracted from the 30m DEM. The maps show that various eroded portions of faults (black arrows) are automatically excluded using the weighted linearity filter and are not included in the analysis.

**Referee 1, Comment 4**

Figure 3: I am a bit surprised by the rose-diagram of fault strike (b). There is this big NW trend coherent with the main orientation of the Afar rift. The second WSW to SW trend surprises me. If you look at the southern Afar (Karrayyu, Adda'Do) on the map, the faults rather seem to trend SSW. And there is almost nothing in this sector of your rose diagram. This effect is even stronger on the rose diagram of figure S4. I'm also surprised you don't have more of the NNW orientations which you can observe on the map in Tat'Ali, Makarassou (and which are more present in the rose diagram of Polun et al., 2018). I did not do any quantitative assessment of it, but it "looks" strange to me. Could you check upon this? Just to make sure there was not a small mistake in the algorithm calculating the azimuths or plotting the rose diagram.

**Response**

As suggested, we checked the code to look for some errors in the strike calculation and plotting but we did not find any mistake in the workflow: our code simply plots strikes using the Python mplstereonet library. The reason behind the apparent visual mismatch between the map and the rose-diagram lies in the nature of the strike calculation. Two approaches are commonly used in literature, the average strike or the tip-to-tip strike. The former method is influenced by the along-fault variation of the fault strike, while the latter can be affected by (sometimes large) changes in the strike induced by secondary structures at the fault tips. In our study, since we cannot control where, approaching the tip, the algorithm should stop tracing the fault, we decided to measure the average strike. This minimizes the impact of secondary structure at the tips and gives more weight to the central portion of the fault. This might also explain the visual difference between mapped faults and rose-diagrams, as also the differences with datasets from manual mapping in Polun et al. (2018). We now clarified this in line 234 as also in the captions of Fig. 3 and Fig. S4 of the revised manuscript.

**Referee 1, Comment 5**

L. 270-272: rates. "The comparison was possible only where well-constrained rock dating was available. Therefore, we used rock ages available close to the profiles (Courtillot et al., 1984; Kidane et al., 2003; Lahitte et al., 2003; Feyissa et al., 2019) but it was not possible to apply this approach to the entire strain map." I don't really understand why you didn't calculate more points. We see only 9 points on fig. 4b. There are much more datings available, from the sources you mention and from others (e.g. Kidane et al., 2003, Courtillot et al., 1984, Manighetti et al., 1998, Zumbo et al., 1995 [I have a shapefile with a compilation (and corrected coordinates) here if you want to check: https://doi.org/10.5281/zenodo.7410073 ]). I guess it would not be difficult for you to calculate geological strain rates for more points where datings are available. These strain rates are very interesting and allow a better comparison with Recent geodetic strain rate, and they would allow to strengthen considerably your interpretation. But maybe there is a good reason why you only restricted this quantification to a few points?

**Response**
Besides the selection of points close to the profiles, the strategy of using a restricted number of data has been aimed at reducing the spatial variability of rock ages, while ensuring a dataset representative of the full evolution of the recent magmatism in Afar and a correct calculation of geological strain rates. The age catalogs show that most nearby samples have similar ages. In this case, we selected those with low uncertainties falling close to the profiles. We also selected only well-constrained data sampled at the top of Upper and Lower Stratoids formations, avoiding older products at the base of fault scarps, and spanning the full range of possible ages across the whole rift. For the most recent volcanic product, we simply selected rock ages representative of the axial volcanism close to the profiles. Other samples have been excluded because the uncertainty was not reported or because the rock age was not compatible with the given formation. Examples of this are several rock ages located on the Upper Stratoids around Manda Inakir but with incompatible ages > 5.4 Ma, as also correctly pointed out in the catalog mentioned by the referee.

For the main tecto-magmatic features in Afar this approach ensured a smooth reconstruction of the strain evolution, but we agree that including more data can strengthen our interpretation of other sectors mentioned in the manuscript, for example Makarassou, Manda Inakir and the Northern sector of Dabbahu. We now included in our analysis 5 additional data from Rime et al. (2023) and Tortelli et al. (2025), reaching a total of 15 rock age constraints. Details about the interpretation and discussion of these new data are reported in the response to comment 7 below. We also included a more detailed explanation on the selection of rock ages at line 286-292 and updated figure 1 and figure 4 with the new data and geological strain rates.

**Referee 1, Comment 6**
L. 426: Where does this 1000-1600 m value come from? Previously, you mentioned 200-1000 m from Abbate et al. (1995).

**Response**
Yes, the thickness of the deposits is from Abbate et al. (1995). We now added the citation as suggested.

**Referee 1, Comment 7**
L. 452-453: "During the last 0.6 Ma, the external sectors of the Afar rift, including Makarassou and MI are not the main locus of strain following the migration and focusing of extension to the rift axis." I think that this conclusion point deserves more discussion, in particular concerning the Makarassou and Manda Inakir region. I was also always intrigued by this area and by this process of strain focusing. In Manda Inakir, the strain is not huge, but it's still clearly faulted. However, the volcanic rocks there are quite Recent. Lahitte et al. (2003) and Manighetti et al. (1998) show age <0.6 Ma, down to 0.03 Ma. So, what is the geological strain rate? I'd guess that with these recent ages, it's not that low. This would mean it was still a locus of strain in the last 0.6 Ma.

Let's take another example: the West Harak graben (12.67N, 41.09E). It is also very recent (0.62-0.07 Ma, Lahitte et al., 2003), heavily faulted and accommodates significant strain according to your results. So, I'd guess that the geological strain rate is also quite high. This suggests, again (maybe even more clearly than MI), that the switch to Dabbahu - Manda Hararo happened more recently than 0.6 Ma.

One could even hypothesize that these peripheral regions became inactive very recently (100s, 1000s, 10000s of years) [but that's a wet-finger guess]. And this would have important implications. It would mean that this change is not necessarily representative of a long-term trend of strain localisation at long-lived and final rift segments, but rather a short-term alternation between different areas of strain accommodation. And that these areas of strain accommodation might have changed in the past and might change again in the future. At larger time and spatial scale, the Afar region has already seen multiple rift axis jumps and abandonments. So, I'm wondering whether what we see here is a (final) rift localisation, or just one rift jump among many others.

Without quantification, this is difficult to answer. And this brings me back to my comment above: why didn't you calculate more geological strain rates points? With more points in Makarassou, Manda Inakir, but also elsewhere in Afar, you could significantly strengthen your point. With only 9 data points, it's hard to be very conclusive.

But I'm probably falling down a rabbit hole, and it's not the main point of the paper. Material for further reflections in the future.

To sum up this comment, I think you should discuss this topic more explicitly in the discussion, taking into account the aspects mentioned above, before presenting it in the conclusion. Or just mention what we know for sure: the MI (and other peripheral regions) currently does not accommodate extension, but we don't know exactly when this change
happened.

**Response**

As suggested by the referee and explained in the response to comment 5, we now included more age constraints on the Manda Inakir (MI) and Makarassou to strengthen our interpretation on these sectors. In particular, we used one rock age from the Upper Statoids in Makarassou (2.3 Ma) and two age constraints for both the Stratoids and the axial products in MI (2.4 Ma and 0.6 Ma, respectively). Furthemore, we also included a new age constraint and a comparison between geological and geodetic strain rates for the northernmost portion of Dabbahu-Manda-Hararo.

At Makarrasou, geological strains are equal to 0.3 where the Upper Stratoids are dated to ~2.3 Ma, resulting in geological strain rates of ~$1.3 \times 10^{-7}$/yr. These values are higher than current geodetic strain rates (< 0.2 x $10^{-7}$/yr) and comparable to those observed elsewhere on the Upper Stratoids. These observations are in agreement with our interpretation of distributed strain during the emplacement of the Upper Stratoids and subsequent focusing of rifting at axial magmatic segments.

At Manda Inakir, geological strains equal to 0.22 are measured on faults cutting the Upper Stratoids (~2.4 Ma), giving geological strain rates of $0.9 \times 10^{-7}$/yr. Conversely, faults forming on lava flow of the axial series (0.6 Ma) accommodated geological strains equal to 0.15, giving a strain rate of $2.5 \times 10^{-7}$/yr, the same as those measured at the axis of AG. However, current geodetic strain rates are way lower and equal to ~$0.2 \times 10^{-7}$/yr. The lack of overlap between more recent rock ages and geological strain measurements prevents us from getting a better estimation of the timing of this strain reduction.

At Dabbahu, geological strains equal to 0.02 have been measured on faults cutting the recent volcanic products (0.024 Ma). This gives the highest geological strain rates of $8.3 \times 10^{-7}$/yr. The same area

shows the highest current geodetic strain rates ($> 3.5 \times 10^{-7}$/yr). We included these new calculations at lines 310-313, 320-324 and 342-344

As correctly pointed out by the referee, high geological strain rates associated with axial volcanism at MI suggest that this magmatic segment has been very active at least at 0.6 Ma. Furthermore, this observation further supports our interpretation that diking and magmatism promote increasing strain rates at magmatic segments. We also agree that our dataset does not allow us to tell if MI is completely abandoned and to estimate the exact timing of deactivation. We clarify this at lines 424-427. However, geological and geodetic strain rates at the rift scale indicate that MI stopped being a main locus of strain sometimes after 0.6 Ma and that rifting now mainly occurs at AG and DMH. Our interpretation is also in agreement with the new insights provided by geochemical analyses and rock dating in Tortelli et al. (2025), that we included in our discussion at line Line 429-431.

**Referee 1, Technical Corrections**
**Figure 4:** A nice-to-have (but not needed) improvement of your figure could be the addition of your calculated geological strain rates (white dots on b) projected on the cross-sections (c-h). It would help to link both datasets and showcase these nice results a bit better. Just a suggestion.
**Response:** We agree and this is something we actually considered during the first writing of the draft. However, including high geological strain rates (up to $6.0 \times 10^{-7}$/yr) would need a wider range of values on the right y-axis and this causes our orange curves to be squeezed to almost straight lines. We thus prefer to keep the figure in its original version.

**Referee 2, Comment 12**
Line 139 and following. I think the Authors should explain in some more detail why they use both 30m- and 90m-resolution DEMs. The choice of a 90m-resolution DEM may sound indeed rather strange, being many 30m-resolution DEMs (e.g., Aster, SRTM) available. This is of course because the Authors want to apply their approach to different datasets with different resolution and compare the results, but I think something more could be said here to explain this. They could also explain, for instance, why the FABDEM has been compared to a 90m-resolution DEM instead of the abovementioned 30m Aster or SRTM DEMs.
**Response**
We applied the method on both 30m and 90m DEMs to test its effectiveness at different spatial resolution. We clarify this at new line 142 of the revised manuscript.

**Referee 2, Comment 15**
Line 237. Not clear why the Authors discuss this difference in terms of 'fault segmentation', whereas in the previous lines they refer to 'faults'. If they plotting faults (and not fault segments) the difference may be simply related to the (obvious) fact that the 90m DEM has a lower resolution and cannot capture minor/shorter faults as the 30m DEM can do (indeed the main difference in the graph are related to faults <5km long). Or did I miss something?
**Response**
Yes, a lower 90m resolution certainly implies that we do not solve smaller faults that we instead detect at 30m. But it can also make two adjacent faults to be solved as a unique longer fault. This would reduce the total number of faults in the dataset. We clarify this at new lines 240-245 of the revised manuscript.

**Referee 2, Comment 19**
Line 268-270. This sentence is not very clear to me. Maybe the Author should first state that strain rates are calculated by use of rock dating (where available), and then extrapolated by assuming the continuity of rifting?
**Response** We modified the sentence as suggested.

**References**

Dumont, S., Klinger, Y., Socquet, A., Doubre, C., and Jacques, E.: Magma influence on propagation of normal faults: Evidence from cumulative slip profiles along Dabbahu-Manda-Hararo rift segment, J. Struct. Geol., 95, 48–59, https://doi.org/10.1016/j.jsg.2016.12.008, 2017.

Geoffroy, L., Le Gall, B., Daoud, M. A., Jalludin, M.: Flip-flop detachment tectonics at nascent passive margins in SE Afar. Journal of the Geological Society, 171 (5), 689–694. doi: https://doi.org/10.1144/jgs2013-135, 2014.

Gupta, A., and Scholz, C. H.: Brittle strain regime transition in the Afar depression: Implications for fault growth and seafloor spreading, *Geology*, 28 (12): 1087–1090. https://doi.org/10.1130/0091-7613(2000)28<1087:BSRTIT>2.0.CO;2, 2000

Hofmann, B., Magee, C., Wright, T.: Throw distribution across the Dabbahu–Manda Hararo dike-induced fault array: Implications for rifting and faulting, Geology, v. 53, p. 161–165, https://doi.org/10.1130/G52665.1, 2024

La Rosa, A., Pagli, C., Keir, D., Sani, F., Corti, G., Wang, H., & Possee, D.: Observing oblique slip during rift linkage in northern Afar. Geophysical Research Letters, 46, 10,782–10,790. https://doi.org/10.1029/2019GL084801, 2019.

Stab, M., Bellahsen, N., Pik, R., Quidelleur, X., Ayalew, D., and Leroy, S.: Modes of rifting in magma-rich settings: Tectono-magmatic evolution of Central Afar, Tectonics, 35, 2–38, https://doi.org/10.1002/2015TC003893, 2016.

Tortelli, G., Gioncada, A., Pagli, C., Barford, D. N., Corti, G., Sani, F., Mark, D. F., Dymock, R. C., Gebru, E. F., and Keir, D. Volcanism records plate thinning driven rift localization in Afar (Ethiopia) since 2-2.5 million years ago. Commun Earth Environ 6, 395, https://doi.org/10.1038/s43247-025-02356-4, 2025